# Novel Concepts in Systemic Sclerosis Pathogenesis: Role for miRNAs

**DOI:** 10.3390/biomedicines9101471

**Published:** 2021-10-14

**Authors:** Iulia Szabo, Laura Muntean, Tania Crisan, Voicu Rednic, Claudia Sirbe, Simona Rednic

**Affiliations:** 1Department of Rheumatology, “Iuliu Hațieganu” University of Medicine and Pharmacy Cluj-Napoca, 400000 Cluj-Napoca, Romania; rotaru.iulia@umfcluj.ro (I.S.); claudia.sirbe@yahoo.com (C.S.); srednic.umfcluj@gmail.com (S.R.); 2Department of Rheumatology, County Emergency Hospital Cluj-Napoca, 400000 Cluj-Napoca, Romania; 3Department of Medical Genetics, “Iuliu Hațieganu” University of Medicine and Pharmacy Cluj-Napoca, 400000 Cluj-Napoca, Romania; tania.crisan@gmail.com; 4Department of Internal Medicine and Radboud Institute for Molecular Life Sciences (RIMLS), Radboud University Medical Center, 6525 GA Nijmegen, The Netherlands; 5Department of Gastroenterology, “Iuliu Hațieganu” University of Medicine and Pharmacy Cluj-Napoca, 400000 Cluj-Napoca, Romania; rednicvoicu@gmail.com; 6Department of Gastroenterology II, “Prof. Dr. Octavian Fodor” Regional Institute of Gastroenterology and Hepatology, 400000 Cluj-Napoca, Romania

**Keywords:** systemic sclerosis, pathogenesis, epigenetic mechanisms, miRNAs

## Abstract

Systemic sclerosis (SSc) is a rare connective tissue disease with heterogeneous clinical phenotypes. It is characterized by the pathogenic triad: microangiopathy, immune dysfunction, and fibrosis. Epigenetic mechanisms modulate gene expression without interfering with the DNA sequence. Epigenetic marks may be reversible and their differential response to external stimuli could explain the protean clinical manifestations of SSc while offering the opportunity of targeted drug development. Small, non-coding RNA sequences (miRNAs) have demonstrated complex interactions between vasculature, immune activation, and extracellular matrices. Distinct miRNA profiles were identified in SSc skin specimens and blood samples containing a wide variety of dysregulated miRNAs. Their target genes are mainly involved in profibrotic pathways, but new lines of evidence also confirm their participation in impaired angiogenesis and aberrant immune responses. Research approaches focusing on earlier stages of the disease and on differential miRNA expression in various tissues could bring novel insights into SSc pathogenesis and validate the clinical utility of miRNAs as biomarkers and therapeutic targets.

## 1. Introduction

Systemic sclerosis (SSc) is a rare autoimmune disease with miscellaneous clinical manifestations and a distinct autoantibody profile [1,2]. It is characterized by high morbidity and mortality related to the extent of fibrosis and obliterative vasculopathy of the internal organs [3,4,5]. The etiology of SSc is not fully unraveled, but evidence supports a complex interaction between genetic variants, environmental exposures, and epigenetic modifications [6]. The modest effect size of SSc-associated genetic risk loci shifted the interest of the scientific community toward the contribution of epigenetics to disease predisposition and its complex pathogenesis [7,8,9].

SSc pathophysiology is distinguished by the interaction between three main altered pathways: microangiopathy, immune dysfunction, and fibrosis [10,11,12]. An inaugural vascular injury [13,14,15] leads to activation of cell-mediated and humoral immune responses [16,17], subsequently resulting in fibroblast to myofibroblast differentiation [18] with production and deposition of collagen and other extracellular matrix (ECM) components into the vascular walls, skin, and internal organs [19,20,21].

Epigenetics refers to the modulation of gene expression through heritable and reversible alterations of the chromatin structure without interfering with the DNA sequence. Epigenetic mechanisms have previously been linked to the pathogenesis of SSc, extensively reviewed elsewhere [22,23,24,25,26]. Evidence of association to SSc has previously been reported for all major epigenetic alterations, including DNA methylation [27,28,29,30,31], histone modifications [32,33,34], non-coding small (miRNA), and long (lncRNA) RNA transcripts [35,36,37]. Epigenetic mechanisms regulate diverse physiological processes such as cell division and differentiation, growth, and development, being responsible at least in part for the variable phenotypic traits in both health and disease [7,9,37]. The epigenome is susceptible to change and can be influenced by various environmental factors, including air pollution, infection, diet, drugs, metals, and chemicals [38,39,40].

DNA methylation is an enzyme-mediated process occurring mostly at the CpG sites where cytosine is located in the vicinity of guanidine in the nucleotide sequence of the DNA structure. DNA methyltransferases (DNMTs) catalyze the addition of a methyl (CH3) group to the 5-carbon of the cytosine ring, generating 5-methylcytosine (5-mC). The methylation status (5-mC content) of a CpG island (cluster of CpG sites) in the promoter region of a gene modulates gene transcription. This translates into either gene-silencing if highly methylated or active gene transcription in low methylated states [24,36,41]. Histone modifications refer to post-translational alterations (such as methylation, acetylation, phosphorylation, ubiquitylation, or sumoylation) of the histone proteins, which alter their interaction with the DNA strand. The subsequent conformational changes in the chromatin architecture make the DNA more or less accessible to transcriptional factors, resulting in activation or repression of gene transcription [42,43]. Non-coding RNAs, miRNAs (<30 nucleotides) [44,45], and lncRNAs (>200 nucleotides) [46,47] are functional regulators of gene expression at the transcriptional and post-transcriptional level. These RNA fragments are transcribed from the DNA but are not translated into proteins [22,23,26,48].

MiRNAs, the focus of this review, bind post-transcriptionally to a complementary sequence from a target mRNA and induce gene silencing. This can be achieved by blocking mRNA translation or promoting mRNA cleavage based on the degree of complementarity [24,37]. MiRNAs have the ability of regulating multiple mRNA targets, whereas translation of one mRNA transcript into protein can be modulated by various miRNAs [49,50] (Figure 1).

Upregulation or downregulation of diverse miRNAs has been identified in blood samples and tissue biopsies from patients with SSc [34,51]. MiRNAs involved in fibrosis received particular attention compared to the scarce data on immune disfunction and vasculopathy [52]. An even more attractive aspect besides a better understanding of the contribution to disease pathogenesis is their potential use as diagnostic and prognostic markers as well as the possibility of developing targeted therapies [53].

The purpose of this review is to illustrate the current knowledge on the role of miRNAs in modulating the three main pathogenic pathways in SSc as well as depicting their clinical utility as biomarkers and therapeutic targets.

## 2. Serum- and Tissue-Specific miRNA Signatures in SSc

Multiple studies have aimed at identifying miRNAs involved in the pathogenesis of SSc and their potential as diagnostic or prognostic biomarkers, as well as therapeutic targets. In this regard, Zhu H (2012) identified a plethora of miRNAs differentially expressed in SSc skin biopsies compared to healthy controls (HC). The miRNA profiles differed between the limited (lcSSc) and diffuse (dcSSc) clinical subtypes. Twenty-one miRNAs overlapped between the two SSc subgroups, out of which six (miR-21, miR-31, miR-503, miR-146, miR-29b, miR-145) were predicted to target mRNAs involved in fibrosis. Further, the analysis was restricted to the TGF-β-associated genes and the miRNAs that regulate their expression levels in both skin specimens and SSc fibroblasts: SMAD7 (miR-21 predicted target), SMAD3 (miR-145 predicted target), and COL1A1 (miR-29b predicted target). In these samples, miR-21 increased levels were mirrored by SMAD7 downregulation, whereas miR-145 and miR29b decreased levels were associated with SMAD3 and COL1A1 upregulation. Stimulation of healthy dermal fibroblasts with recombinant TGF-β resulted in increased miR-21/decreased SMAD7, increased miR-145/decreased SMAD3, and decreased miR-29b/increased COL1A1 levels, suggesting that these miRNAs do not directly control their target mRNAs [34].

Interestingly, these miRNAs were not reproduced in the study conducted by Li (2012). By means of miRNA microarray analysis, 24 miRNAs were identified as being differentially expressed in SSc skin samples. Results were confirmed by real-time PCR. Target genes with a known role in SSc pathogenesis were identified for six miRNAs (hsa-miR-206, hsa-miR-133a, hsa-miR-125b, hsa-miR-140-5p, hsa-miR-23b, hsa-let-7g) using bioinformatics analysis. Hsa-miR-206 received particular attention as it regulates an impressive number of genes, 15 of them being correlated with SSc pathogenesis [54].

As expected, miRNAs identified in SSc serum samples differ from tissue miRNAs. Steen (2015) proposed a circulating miRNA signature in a large cohort of 189 patients. The study included 120 SSc patients, 29 systemic lupus erythematosus (SLE) patients, and 40 HC. From the 37 identified miRNAs, 19 were significantly dysregulated (14 miRNAs decreased and five miRNAs increased). Quantitative PCR reflected the main differences between SSc patients and HC with respect to the expression of the miRNA 17~92 cluster, as well as miR-16, miR-223, and miR-638. Predicted targets of these miRNAs are mRNAs involved in different fibrotic pathways, including TGF-β [51].

A different miRNA circulating profile was demonstrated by means of microarray analysis in the serum of 10 SSc patients compared to six HC in a study by Rusek et al. (2019). Out of the 15 miRNAs differentially expressed, miR-4484 was remarkably increased (18-fold). Bioinformatics analysis suggested miR-4484 as a potential regulator of fibrosis through the identification of a wide range of target genes involved in the TGF-β/SMAD and Wnt/β-catenin signaling pathways, as well as collagen synthesis and extracellular matrix (ECM) homeostasis [45]. Matrix metalloproteinase-21 (MMP-21), even though not a direct target gene according to computational analysis, was hypothesized to be up-regulated by miR-4484 due to their close chromosomal vicinity and the increased MMP-21 serum levels. These findings further enabled the authors to suggest that miR-4484 and MMP-21 might play a role in SSc pathogenesis and proposed them as serum biomarkers [45].

Another relevant aspect is that circulating miRNA profiles are able to discriminate between SSc clinical subtypes (lcSSc versus dcSSc) and autoantibody specificities, as shown by Wuttge (2015). Out of 45 selected miRNAs, four miRNAs (miR-223, miR-181b, miR-342-3p, miR-184) consistently exhibited different expression levels in the lcSSc and dcSSc subgroups. In the autoantibody subgroups, five miRNAs (miR-409, miR-184, miR-92a, miR-29a, miR-101) showed statistically different expression levels [55].

A distinct miRNA signature in SSc and idiopathic pulmonary fibrosis (IPF) lung fibroblasts was expressed in the experiment led by Mullenbrock (2018). The author proved that various miRNAs were differentially expressed compared to controls. To validate their function, transfection of miR-29b-3p, miR-138-5p, and miR-146b-5p mimics was performed and their effects on gene expression were quantified using a Nanostring fibrosis panel. One hundred seventy-five pro-fibrotic target genes were consequently downregulated in the SSc and IPF lung fibroblasts, supporting a role for miR-29b-3p, miR-138-5p, and miR-146b-5p in fibrosis in these disease models [56].

## 3. MiRNAs: Culprits in SSc Pathogenesis

### 3.1. Profibrotic miRNA Transcripts

Zhu H (2012) identified altered expression levels of miR-21, miR-145, and miR-29b in SSc skin and cultured fibroblasts [34]. A further study from the same group explored the expression levels of miR-21 and its target gene Smad7 in SSc and bleomycin-treated mice skin biopsies. They validated miR-21 as an important regulator of the TGF-β signaling pathway through the manipulation of its direct target, SMAD7. On the one hand, TGF-β fibroblast stimulation induced upregulation of miR-21 and downregulation of Smad7, and on the other hand, transfection of small interfering RNA (siRNA) decreased Smad7 protein levels. Smad7 is a negative regulatory component of the TGF-β signaling pathway. Therefore, decreased levels of Smad7 will have the opposite effect by stimulating fibrosis. Similar results were obtained in the bleomycin-treated mice with upregulation of miR-21 and downregulation of Smad7. After treatment with bortezomib, miR-21 decreased, Smad7 levels were restored, and skin fibrosis improved [57]. The same profibrotic phenotype of miR-21 was demonstrated by S. Jafarinejad-Farsangi (2019). Upregulation of miR-21 was observed in both diffuse cutaneous SSc (dcSSc) and TGF-β-stimulated fibroblasts, leading to increased type I collagen production [58].

Ly (2020) has recently proven that miR-145 mediates α-smooth muscle actin (α-SMA) myofibroblast differentiation through downregulation of transcription factor Kruppel-like factor 4 (KLF4) in TGF-β1-stimulated dermal fibroblasts and SSc fibroblasts. KLF4 has a prohibitory effect on the XYLT1 gene. XYLT1 encodes xylosyltransferase-1 (XT-1), a proteoglycan synthesis biomarker. Experiments revealed that exogenous delivery of KLF4 siRNA into normal human fibroblasts led to downregulation of KLF4 mRNA levels and upregulation of XYLT1 expression levels in a dose-dependent manner in response to TGF-β1. The same trend was identified in SSc fibroblasts, therefore leading to the identification of a new miR-145/KLF4 profibrotic pathway [59].

Another validated profibrotic miRNA is miR-92a. Transfection of miR-92a mimics in normal fibroblasts resulted in decreased expression levels of matrix metalloproteinase-1 (MMP-1). MiR-92a upregulation in SSc fibroblasts and serum from SSc patients might be a consequence of TGF-β endogenous activation as increased miR-92a levels were evidenced in normal dermal fibroblasts stimulated with TGF-β and decreased expression levels were shown after inhibition of TGF-β with siRNA [60]. MMP-1 is also the target gene for another profibrotic miRNA, miR-202-3p, as shown by Zhou (2017). In SSc skin samples and cultured fibroblasts, miR-202-3p was upregulated and MMP-1 was downregulated. Luciferase reporter assays identified MMP-1 as the target gene for miR-202-3p and gain and loss of function assays showed that in SSc fibroblasts MMP-1 was regulated by miR-202-3p [61].

Nakayama (2017) showed that miR-4458 plays a decisive role in type I collagen production via the IL-23 immune pathway, therefore indicating IL-23 as an important factor in SSc fibrogenesis and a possible therapeutic target. In normal fibroblasts, IL-23 stimulation leads to increased miR-4458 levels and downregulation of type I collagen production. Conversely, IL-23 stimulation of SSc fibroblasts also prompts miR-4458 upregulation, but the effect at the protein level is enhanced type I collagen synthesis [62].

MiR-155 also proved to play a role in SSc fibrogenesis by regulating Wnt/β catenin and Akt profibrotic pathways. This finding was illustrated after transfection of mouse fibroblasts with miR-155 inhibitor, which resulted in increased degradation of β-catenin, decreased phosphorylation of Akt, and, subsequently, decreased type I collagen production. In bleomycin-treated miR-155 knockout mice and after topical administration of antagomiR-155 in bleomycin-induced fibrosis mouse models, decreased protein levels of β-catenin and pAkt were evidenced. Additionally, improvement of skin fibrosis was noted, therefore supporting the therapeutic potential of miR-155 inhibition [63]. Christmann (2016) further suggested miR-155 as a potential therapeutic target since miR-155 knockout mice exhibited less aggressive lung involvement and better survival rates after bleomycin administration compared to wild-type controls [64]. Additionally, the same group suggested a promising role for miR-155 as a prognostic biomarker in SSc-ILD due to its correlation with higher high-resolution computed tomography (HRCT) fibrosis scores and lower performances on pulmonary function tests (PFTs) [64].

Data from Artlett (2017) showed that miR-155 expression levels depend upon inflammasome activation. The study depicted the strong link between inflammasome activation, miR-155 expression, and collagen synthesis in SSc fibroblasts and bleomycin mouse models. Inflammasome inhibition in SSc fibroblasts via caspase-1 inhibitor determined downregulation of miR-155 and decreased collagen production. Fibroblasts from NLRP3 knockout mice did not exhibit enhanced miR-155 expression levels after stimulation with bleomycin, showing that miR-155 expression cannot be achieved without inflammasome activation [65].

The study by Henderson (2021) validated miR27a-3p as a profibrotic epigenetic direct regulator of the sFRP-1 protein, a Wnt pathway antagonist. Transfection of miR27a-3p mimic in TGF-β1-stimulated normal dermal fibroblasts induced COL1A1 and Axin-2 upregulation, as well as downregulation of the antifibrotic PPARγ mRNA and decline in MMP-1 protein levels. The authors also revealed decreased sFRP-1 protein levels in the serum and skin biopsies of early dcSSc patients and increased miR27a-3p expression levels in SSc dermal fibroblasts. A 33% drop in collagen synthesis resulted following exogenous delivery of antagomiR27a-3p in sFRP-1-depleted SSc dermal fibroblasts. These results suggest a role for miR27a-3p in SSc fibrosis [66]. Another study from the same group confirmed miR33a-3p as an additional epigenetic regulator of the Wnt pathway through direct repression of Dickkopf-1 (DKK-1) mRNA translation. MiR33a-3p was increased in SSc fibroblasts, whereas DKK-1 was decreased. AntagomiR33a-3p transfection into SSc fibroblasts led to a significant reduction in collagen 1 synthesis, again supporting a profibrotic role for this miRNA in SSc pathogenesis [67].

MiR-483-5p displayed a profibrotic phenotype in SSc. Serum levels of miR-483-5p are elevated in such patients. Transfection of miR-483-5p mimics in primary human fibroblasts and pulmonary endothelial cells caused increased synthesis of type IV collagen via modulation of COL4A1 and COL4A2 target genes. Transfection of miR-483-5p in endothelial cells also increased the expression levels of αSMA and SM22A mRNA, suggesting that miR-483-5p orchestrates the myofibroblast differentiation of endothelial cells [44].

Table 1 summarizes the main profibrotic miRNAs identified so far along with their targeted genes.

### 3.2. Antifibrotic miRNA Transcripts

TGF-β, a promoter of collagen synthesis and fibroblast proliferation and differentiation, plays a central role in SSc pathogenesis [68]. TGF-β signaling is mediated through its receptors, TGF-β receptor type 1 (TGFBR1) and type 2 (TGFBR2) [69]. Numerous in vitro and in vivo experiments have shown that TGFBR2 is involved in dermal and internal organ fibrosis [70,71,72]. From that perspective, Shi (2018) has demonstrated that TGFBR2 upregulation in SSc dermal fibroblasts and in dermal biopsies is a direct consequence of miR-3606-3p downregulation. Additionally, transfection of miR-3606-3p mimics in SSc dermal fibroblasts resulted in a reduction of TGFBR2 expression, as well as reduced p-SMAD2/3 and type I collagen protein levels [73]. MiR-3606-3p silencing of the TGFBR2 mRNA could represent a new therapeutic strategy in SSc.

Besides the profibrotic phenotype displayed by miR-4458, Nakayama (2017) likewise showed that miR-18a influences type I collagen production. In normal fibroblasts, IL-23 stimulation led to decreased miR-18a expression levels and downregulation of type I collagen synthesis. In contrast, IL-23 stimulation of SSc fibroblasts caused miR-18a downregulation and increased type I collagen synthesis. This paradox is explained by strong downregulation of miR-18a, a potent antifibrotic miRNA, due to intrinsic activation of TGF-β in SSc fibroblasts. The profibrotic activity of IL-23 was subsequently demonstrated by accelerated skin fibrosis after IL-23 injection of bleomycin-treated mice [62].

Five members of the let-7 family were dysregulated in SSc and localized scleroderma (LSc) skin samples compared to normal controls and keloid skin specimens. Let-7a was significantly downregulated in scleroderma tissues, with lower levels in the LSc group compared to the SSc group. TGF-β1 stimulation of normal fibroblasts resulted in decreased expression levels of Let-7a and increased production of type I collagen, suggesting that downregulation of Let-7a might mitigate the overexpression of extracellular matrices, mainly the secretion of type I collagen. This hypothesis was validated by transfection of the Let-7a inhibitor in the normal fibroblasts, which led to increased production of type I collagen. Serum levels of Let-7a were also downregulated and the same trend of lower levels in the LSc subset compared to the SSc subset was maintained. Injection of Let-7a in bleomycin-induced fibrosis mouse models resulted in improvement of skin fibrosis [74].

Maurer (2010) demonstrated significant downregulation of miR-29a in SSc-cultured fibroblasts, SSc skin biopsies, and bleomycin-induced fibrosis mouse models. In order to validate its function and role in SSc fibrogenesis, transfection of pre-miR-29a/29b/29c in SSc fibroblasts was conducted. This manipulation led to downregulation of type I collagen and markedly decreased expression levels of type III collagen being observed after pre-miR-29a transfection. Conversely, transfection of anti-miR-29a in normal fibroblasts determined upregulation of type I and type III collagens. COL3A1 proved to be a direct target of miR-29a after cotransfection of HEK 293 cells with pre-miR-29a and pGL3 luciferase reporter containing the 3’-UTR of COL3A1. Cotransfection resulted in reduced relative luciferase activity, whereas cotransfection with anti-miR-29a and pGL3 luciferase reporter led to enhanced relative luciferase activity. The group subsequently analyzed the influence of several profibrotic cytokines, namely TGF-β, PDGF-B, and IL-4, on miR-29a expression. They demonstrated that stimulation of normal fibroblasts with these molecules resulted in downregulation of miR-29a similar to levels seen in SSc fibroblasts, whereas inhibition of TGF-β and PDGF-B pathways with imatinib restored miR-29a levels in SSc fibroblasts as well as bleomycin-induced skin fibrosis. Given the direct regulation of collagen genes by miR-29a, this miRNA could be a potential antifibrotic therapeutic target [75]. In a recent study by Jafarinejad-Farsangi (2019), transfection of miR-29a mimics significantly reduced collagen type I expression levels in SSc and TGF-β-stimulated fibroblasts, further supporting the antifibrotic role of miR-29a [58].

Similarly, Ciechomska (2014) validated TAB1 as another target gene for miR-29a, demonstrating an important role for this miRNA in SSc fibrosis. Transfection of miR-29a in normal fibroblasts led to downregulation of TIMP-1 and upregulation of MMP-1, resulting in decreased extracellular matrix deposition. Bioinformatics analysis identified TAB1 as a possible target gene for miR-29a. Validation of TAB1 was performed through cotransfection of HeLa cells with pre-miR-29a and TAB1 3’UTR luciferase reporter. Luciferase analysis showed a 20% reduction in luciferase activity after cotransfection. Subsequently, pharmacological inhibition of TBA1 or transfection of anti-TAB1 siRNA in normal fibroblasts resulted in TIMP-1 reduction, demonstrating that TAB1 plays a key role in the regulation of TIMP-1 expression levels [76].

Honda (2013) depicted miR-150 as an antifibrotic miRNA that mediates its effects via integrin β3 inhibition. Integrin β3 is an adhesion molecule that is supposed to play an important role in the endogenous TGF-β activation in SSc fibroblasts. In SSc skin and cultured fibroblasts, low miR-150 levels and high integrin β3 levels were identified. Transfection of miR-150 mimics in SSc fibroblasts resulted in decreased integrin β3, phosphorylated SMAD3, and type I collagen, while on the contrary miR-150 antisense inhibition in normal fibroblasts caused enhanced expression of the aforementioned molecules [77].

PDGF receptor β is the target gene for miR-30b. Tanaka (2013) demonstrated that miR-30b was repressed in SSc serum samples. Decreased levels were also seen in SSc skin specimens and experimental mouse models, whereas PDGFR-β was highly expressed in SSc fibroblasts compared to controls. Hence, downregulation of miR-30b leads to a profibrotic phenotype via enhanced expression of the PDGFR-β [78].

MiR-135b and miR-196a are validated antifibrotic miRNAs. O’Reilly (2016) proved that IL-13 signaling leads to increased extracellular matrix deposition in SSc fibroblasts through regulation of the signal transducer and activator of transcription-6 (STAT6). IL-13-induced downregulation of miR-135b results in upregulation of STAT6 and increased collagen synthesis [79]. The involvement of epigenetics in SSc fibrosis is also illustrated by Makino (2013). Regulation of discoidin domain receptor 2 (DDR2) mRNA and protein level is accomplished through negative feedback: decreased DDR2 stimulates miR-196a expression and decreased collagen synthesis in normal fibroblasts. In SSc fibroblasts, this feedback is incompetent due to downregulation of miR-196a by endogenous activation and downstream signaling of TGF-β, generating enhanced collagen production [80].

MiR-125b modulates both the activation of fibroblasts into myofibroblasts and fibroblast apoptosis. It exerts a tissue dependent effect as seen with cancer and cardiac fibrosis [81,82]. Kozlova (2019) demonstrated that miR-125b is downregulated in SSc dermal fibroblasts and skin samples. This leads to enhanced fibroblast apoptosis through induction of apoptosis genes BAK1, BMF, and BBC3, but also reduces fibroblast proliferation and differentiation as shown by decreased αSMA mRNA expression and protein levels. Hence, miR-125b plays a protective, antifibrotic role in SSc pathogenesis [83].

MiR-16-5p inhibits tissue fibrosis by repressing myofibroblast activation through direct inhibition of NOTCH2 expression. Yao (2020) revealed that transfection of antogomiR-16-5p in cultured skin fibroblasts led to a rise in the levels of several profibrotic markers, such as COL1A1, COL1A2, connective tissue growth factor (CTGF), as well as α-SMA, a marker of myofibroblast differentiation. On the contrary, MMP-1 and matrix metalloproteinase-8 (MMP-8) levels were decreased in response to miR-16-5p inhibition. Additional exogenous delivery of siNOTCH2 partially reversed the expression of the abovementioned biomarkers. Decreased miR-16-5p and increased NOTCH2 expression levels were identified in SSc serum samples, suggesting that miR-16-5p interferes in SSc pathogenesis by modulating fibroblast to myofibroblast differentiation [84].

Table 2 outlines the main characteristics of the antifibrotic miRNAs identified in patients with SSc.

Figure 2 is an illustration of the regulatory effects of various profibrotic and antifibrotic miRNAs involved in SSc tissue fibrosis.

### 3.3. Apoptosis and miRNAs

SSc fibroblasts are resistant to apoptosis. This dysfunctional programmed cell death further contributes to increased extracellular matrix deposition [85]. Two members of the Bcl-2 family, namely Bax and Bcl-2, control apoptosis. Mir-29a and miR-21 regulate the expression levels of Bax and Bcl-2 [86,87].

Accordingly, Jafarinejad-Farsangi (2015) demonstrated the proapoptotic role of miR-29a in SSc and TGF-β-stimulated fibroblasts through regulation of the expression levels of the Bcl-2 family members. MiR-29a downregulates the antiapoptotic Bcl-2 and Bcl-XL proteins, therefore increasing the Bax:Bcl-2 ratio. An elevated Bax:Bcl-2 ratio translates into enhanced apoptosis. The dual properties of mir-29a, both antifibrotic and proapoptotic, make it an excellent therapeutic target [86].

In a subsequent study, Jafarinejad-Farsangi (2016) confirmed the increased Bcl-2 levels in SSc fibroblasts and the resulting decreased Bax:Bcl-2 ratio. This confers resistance to apoptosis, as previously demonstrated [86]. Transfection of SSc fibroblasts with miR-21 mimics additionally upregulated Bcl-2 levels and lowered the Bax:Bcl-2 ratio, supporting miR-21 as an antiapoptotic factor. On the contrary, transfection of miR-21 inhibitor increased Bax expression levels and consequently enhanced apoptosis. MiR-21 inhibition is an attractive therapeutic target in inducing apoptosis and reversing fibrosis in SSc [87].

Table 3 is a schematic representation of the miRNAs that modulate apoptosis in SSc.

### 3.4. Microangiopathy and miRNAs

Proliferative microangiopathy is responsible for severe manifestations such as digital ulcers and pulmonary arterial hypertension. Iwamoto (2016) investigated the potential role of epigenetics in mediating SSc vasculopathy. The study revealed that SSc fibroblasts and dermal biopsies exhibited lower levels of miR-193b compared to normal controls. By means of computational analysis, several genes were identified as potential targets and urokinase-type plasminogen activator (uPA) was the most significantly dysregulated by miR-193b stimulation or inhibition. Accordingly, transfection of miR-193b mimics lowered uPA levels and transfection of miR-193b inhibitors upregulated uPA expression. These findings were also shown in HC and primary human pulmonary artery smooth muscle cell (HPASMCs) cultures and validated at the protein level. Immunohistochemistry and double-staining of SSc skin samples with uPA and α-SMA proved that uPA is highly expressed in vascular structures, especially by vascular smooth muscle cells (VSMCs), and modestly expressed by skin fibroblasts. Obtained data suggest a possible role of uPA in SSc vasculopathy that was further validated on HPASMCs cultures where uPA enhanced the expression of the PCNA proliferation marker and decreased apoptosis detected by flow cytometry. Upregulation of MiR-193b represents a potential treatment for targeting vasculopathy in SSc [88].

MiR-126 is a negative regulator of epidermal growth factor like-domain 7 (EGFL7), a modulator of angiogenesis that exhibits pro-angiogenic properties. According to the publication of Liakouli (2019), EGFL7 expression levels are increased in early onset dcSSc skin specimens but decreased in long-standing dcSSc skin biopsies. The authors further showed that exogenous delivery of human recombinant (rh)EGFL7 suppressed the impaired angiogenesis in cocultures of early-onset and long-standing dcSSc fibroblasts with HUVECs. Moreover, (rh)EGFL7 suppressed COL1A1 expression levels in early-onset SSc fibroblasts, whereas EGFL7 small interfering (si)RNA increased COL1A1 mRNA levels. These results emphasize the dual role of EGFL7 in SSc pathogenesis, modulating both angiogenesis and fibrosis [89].

Table 4 represents a summary of the modulatory effects of miRNAs in SSc vasculopathy.

### 3.5. Immune Dysfunction and miRNAs

B cell-activating factor (BAFF), a TNF superfamily member, revealed its important role in the pathogenesis of several autoimmune diseases by modulating the activity and survival of B cells. In SSc, stimulation of dermal fibroblasts with either Poly(I:C) or IFN-γ (known upregulators of BAFF) resulted in decreased expression levels of miR-30a-3p. Conversely, transfection of miR-30a-3p mimics in these cells lowered BAFF expression levels and consequently determineed decreased B cell survival. To a further extent, transfection of normal fibroblasts with miR-30a-3p inhibitor enhanced BAFF levels, demonstrating that miR-30a-3p is an important regulator of BAFF production and secretion [90].

The interplay between miRNA dysregulation and interferon (IFN) signatures in SSc was explored by Ciechomska (2020) through mRNA–miRNA sequencing and functional studies on monocytes. Accordingly, miR-26a-2-3p was significantly downregulated in SSc monocytes compared to controls, while expression of selected IFN-stimulated genes was increased in SSc monocytes but not in controls or rheumatoid arthritis samples. Transfection of miR-26a-2-3p mimics to TLR-stimulated THP-1 cells proved that this miRNA is a negative regulator of IFN-stimulated genes. These findings suggest that miR-26a-2-3p downregulation might be responsible, at least in part, for the increased IFN production in SSc [91].

Table 5 illustrates the characteristics of miRNAs involved in the dysregulation of the immune system in SSc.

## 4. MiRNAs: Diagnostic and Prognostic Biomarkers

An interesting finding was that of Makino (2012), who found highly expressed levels of miR-142-3p in the serum of SSc patients. These levels were significantly dysregulated compared to the scleroderma spectrum disorder (SSD), systemic lupus erythematosus (SLE), and dermatomyositis (DM) cohorts, suggesting that miR-142-3p could be a potential diagnostic marker in distinguishing SSc from SSD [92].

Izumiya (2015) illustrated the relationship between five let-7 family miRNA members and the severity of pulmonary hypertension in SSc patients. Microarray analysis of skin biopsies from six patients without pulmonary hypertension (PH) and nine patients with PH identified 32 miRNAs that were upregulated and 14 miRNAs that were downregulated. After validation by quantitative real-time PCR, the expression levels of let-7a, let-7d, let-7e, let-7f, and let-7g were significantly dysregulated in the PH group. Furthermore, let-7d and let-7b were correlated with an increased pulmonary arterial pressure measured by echocardiography, making them possible candidates as biomarkers of PH severity in SSc patients [93].

The association between cancer and SSc is another troubling aspect in the management of these patients. SSc patients have a higher risk of developing certain types of cancer, mainly breast, lung, and hematological malignancies [94]. Dolcino (2018) investigated the potential role of epigenetics in promoting carcinogenesis in SSc. The expression levels of 5 MiRNAs (miR-21-5p, miR-92a-3p, miR-155-5p, miR-16-5p, miR-126) with proven implication in these types of malignancies were detected by real-time PCR in the serum of 30 SSc patients and 10 HC. MiR-21-5p, miR-92a-3p, miR-155-5p, and miR-16-5p were significantly dysregulated in the SSc group compared to controls. Mir-126 levels were not statistically different between SSc patients and controls. The upregulation of miR-21-5p, miR-92a-3p, and miR-155-5p in both SSc and cancer specimens, with implications in fibrosis as well as angiogenesis and proliferation, suggests that there might be a defining role for epigenetic mechanisms in cancer predisposition in SSc [95].

## 5. Role of miRNAs in SSc Interstitial Lung Disease (SSc-ILD) Pathogenesis

Pulmonary involvement in SSc is associated with increased morbidity and mortality and therefore warrants special attention with respect to the role miRNAs might play in lung fibrosis.

Wu (2021) analyzed one miRNA and three mRNA datasets retrieved from the Gene Expression Omnibus (GEO) database and identified nine differentially expressed miRNAs in SSc-ILD lung samples compared to controls. These miRNAs regulate various fibrosis-related signaling pathways, such as the integrin family, TNF-related apoptosis inducing ligand (TRAIL) protein, and vascular endothelial growth factor (VEGF)/VEGF receptor (VEGFR) signaling networks [96].

Compared to idiopathic pulmonary fibrosis (IPF), an organ-specific fibrotic disease, Mullenbrock (2018) identified a similar miRNA profile in SSc lung fibroblasts. Transfection of miR-29b-3p, miR-138-5p, and miR-146b-5p in both IPF and SSc pulmonary fibroblasts resulted in downregulation of several profibrotic genes, COL1A1 (miR-29b-3p target gene), connective tissue growth factor (CTGF; miR138-5p target gene), and actin alpha 2 (ACTA2; miR-146b-5p target gene) [97].

As previously mentioned, data from Christmann (2016) promoted miR-155 as an attractive therapeutic target and also a promising prognostic biomarker in SSc-ILD [64]. Another proposed prognostic biomarker is miR-200c identified in peripheral blood mononuclear cells (PBMCs) among patients with ILD and different connective tissue diseases (CTDs). Higher miR-200c levels were detected among patients with SSc-ILD compared to other CTDs and in patients with more severe forms of lung fibrosis (defined by the decline of FVC and FEV1) [98]. Results from a different study showed that transfection of miR-30c in experimental mouse models resulted in decreased dermal thickness and collagen production as well as improved vascular dysfunction and lung fibrosis scores, promoting miR-30c as a versatile therapeutic target in SSc [99].

MiR-320a is downregulated in serum and PBMCs of SSc-ILD patients and lung samples of bleomycin-induced ILD. Through its target genes, TGFR2 and insulin-like growth factor receptor 1 (IGF1R), miR-320a modulates the expression of type I collagen in normal human pulmonary fibroblasts cell lines. Further stimulation of these cells with TGF-β upregulated both miR-320a and collagen genes, again pointing toward the central role of the TGF-β signaling pathway in tissue fibrosis [100].

Pulmonary endothelial myofibroblast differentiation and type IV collagen synthesis are induced in vitro by miR-483-5p, a profibrotic miRNA that was detected in high levels in SSc serum samples [44].

## 6. Future Directions

Research in the field of miRNAs in SSc has mostly focused on miRNAs exhibiting an antifibrotic or profibrotic effect in the hope of identifying and developing more targeted therapies. Some of the earliest promising results came from Montgomery (2014). In this study, bleomycin-induced pulmonary fibrosis improved after intravenous administration of double-stranded miR-29b mimics. These chemically modified miRNA transcripts were able to restore COL1A1 and COL3A1 expression levels and even decrease total collagen amount in lung biopsies. These findings suggest that miR-29b therapeutic delivery may not only stop progression of pulmonary fibrosis but also reverse already established lung fibrosis [101]. In this respect, miRagen Therapeutics has a phase 2, double-blind, placebo-controlled clinical trial investigating the potential use of Remlarsen/MRG-201 (miR-29 mimic) for the treatment of keloid scars (www.clinicaltrials.gov, accessed on 27 March 2021). MRG-229, a second-generation miR-29 mimic designed for treatment of idiopathic pulmonary fibrosis (IPF), has recently shown favorable efficacy and safety profiles in preclinical studies (www.miragen.com, accessed on 27 March 2021). Another attractive target could be the inhibition of miR-155. Several lines of research showed that bleomycin-treated miR-155 knockout mice achieved improved skin and pulmonary fibrosis scores [63,64]. Therefore, silencing profibrotic miRNAs with synthetic antagomiRs could also represent an approach in SSc therapy. Besides miRNA mimics and antagomiRs, several other methods of delivery have been developed, but the most important questions remain their stability, cell and tissue specificity, and subsequent immune response [24,101,102].

Exosomes, small membrane vesicles containing genetic information, are emerging as a new direction in the study of SSc pathogenesis. They mediate intercellular interactions within the same tissue but also modulate cell phenotypes away from their origin in distant organs. Thus, free-circulating exosomes could explain the progression of fibrosis from skin to different organs [103,104]. Current evidence even implies that exosomes might represent the link between the three disrupted mechanistic pathways in SSc: microangiopathy, immune disfunction, and fibrosis [105]. The serum exosome content of 28 miRNAs previously shown to mediate various fibrotic pathways in SSc was evaluated by means of semiquantitative real-time PCR (RT-PCR) in three lcSSc patients, three dcSSc patients, and HC. The expression levels of six profibrotic miRNAs were increased and 10 antifibrotic miRNAs were decreased in both SSc subsets compared to normal controls. A significant difference was also observed in the expression levels of eight antifibrotic miRNAs (miR-let-7a, miR-290, miR-92a, miR-1250, miR-133, miR-140, miR-146a, miR-200a) that were markedly downregulated in the dcSSc subgroup compared to levels observed in the lcSSc subgroup. Furthermore, normal dermal fibroblasts were exposed in vitro to three different concentrations of exosomes isolated from both SSc subgroups in order to validate their involvement in fibrosis. RT-PCR evidenced dose-dependent upregulation of COL1A1, COL3A1, and fibronectin 1 (FN1), genes encoding type I collagen, type III collagen, and fibronectin. Their corresponding protein levels were also increased after exosome exposure. Other genes that were induced after exosome treatment were genes involved in myofibroblast activation as well as genes encoding TGF-β and CTGF 56. As a result, exosomes act as potent biological tools that could be used as diagnostic and even prognostic biomarkers, whereas their manipulation as therapy delivery carriers is an exciting perspective in many diseases, including SSc [105].

## 7. Limitations

This is a narrative literature review that focused on the current state of research in the field of epigenetics, particularly miRNAs, and their role in SSc pathogenesis. “Systemic sclerosis”, “pathogenesis”, “epigenetic mechanisms”, and “miRNAs” were the MeSH terms used to select and retrieve information from the National Library of Medicine (PubMed.gov). Original articles, narrative reviews, systematic reviews, and meta-analysis were considered and included in the present study after applying text availability (only full-text articles) and language (only English) filters. This review is therefore prone to the inherited limitations of this research methodology such as selection bias, difficulty in determining complex interactions, and drawing conclusions.

## 8. Conclusions

MiRNAs are involved in various physiological and pathological processes. These molecules have validated their role in modulating vasculopathy, immune responses, and fibrosis in SSc and represent promising therapeutic targets. Even though advances in the field are continuously expanding, certain limitations remain to be addressed in future studies. SSc heterogeneity, small cohorts, permissive inclusion criteria without a clear distinction in terms of disease severity, and status, together with scarce data regarding current treatments, are just some of the culprits responsible for the discordance between reports. MiRNAs are cell- and tissue-specific, therefore their expression is expected to differ between body compartments and internal organs. In this respect, a more balanced research agenda should also be considered given the fact that most protocols investigated miRNA expression levels in skin biopsies and dermal fibroblasts with an emphasis on profibrotic and antifibrotic transcripts.

## Figures and Tables

**Figure 1 biomedicines-09-01471-f001:**
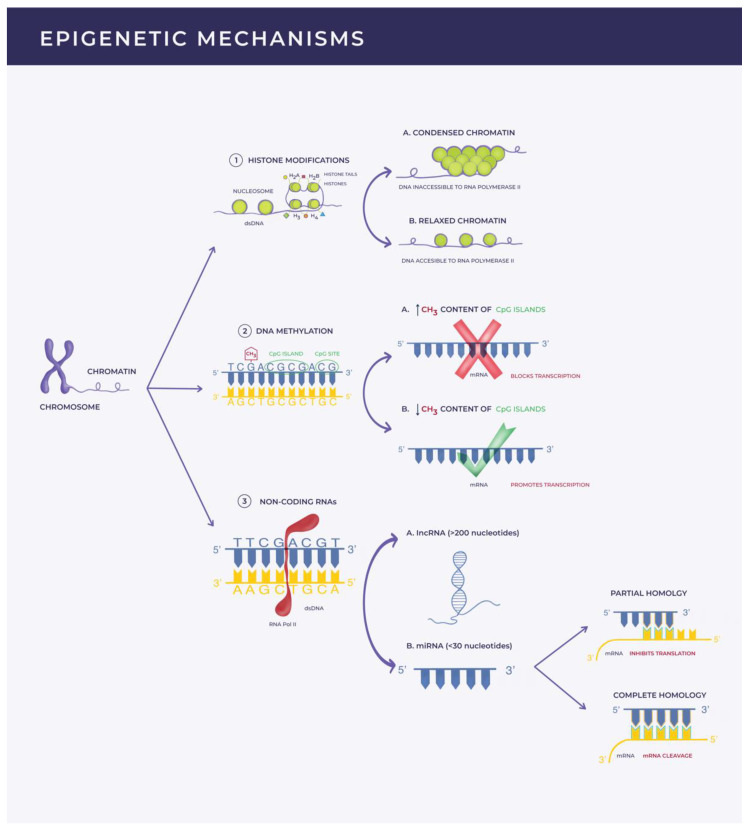
Illustration of epigenetic mechanisms. This figure is a schematical representation of the epigenetic mechanisms that modulate gene expression: (1) *Histone modifications* refer to post-translational modifications of the histone proteins leading to conformational changes that make DNA more or less accessible to RNA polymerase II (RNA POL II); (2) *DNA methylation* is an enzyme-mediated process consisting of the addition of a methyl (CH3) group to the 5-carbon of the cytosine ring from a CpG site. Clusters of CpG sites form a CpG island. The methylation status of a CpG island located in the promotor region of a gene can either lead to gene silencing if highly methylated or active gene transcription if slightly methylated; (3) *Non-coding RNAs* (lncRNAs and miRNAs) are functional RNA fragments transcribed from the DNA by RNA Pol II but unable to be translated into proteins. LncRNAs possess diverse functions, such as the capacity of altering mRNA splicing or recruiting chromatin remodeling proteins and transcription factors. MiRNAs have the ability to bind post-transcriptionally to a complementary sequence from a target mRNA and induce gene silencing. Depending on the degree of homology they can either inhibit transcription or induce mRNA cleavage.

**Figure 2 biomedicines-09-01471-f002:**
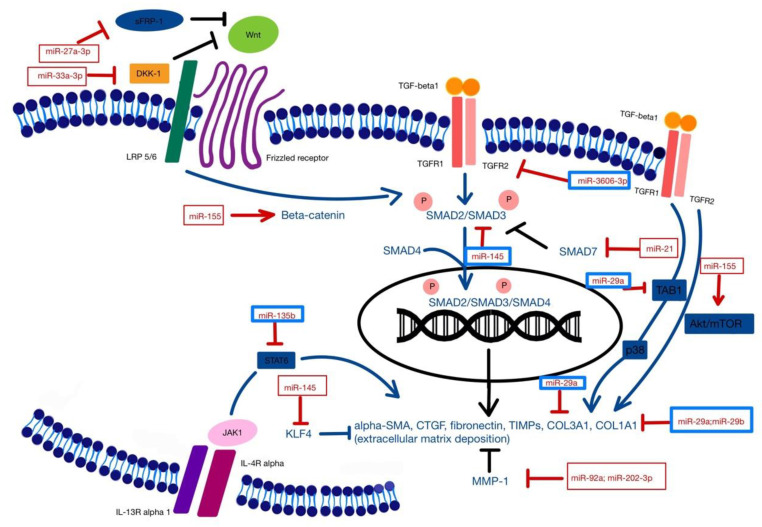
TGF-β1, the main regulator of fibrosis, plays a central role in SSc pathogenesis. SMAD and non-SMAD TGF-β signaling pathways lead to transcription of fibrosis-related genes responsible for fibroblast proliferation, myofibroblast differentiation, and extracellular matrix deposition. Upregulation (red squares) or downregulation (blue squares) of diverse miRNAs interfere with these mechanisms and promote tissue fibrosis. KLF4: Kruppel-like factor 4; MMP1: matrix metalloproteinase 1; sFRP-1: secreted frizzled-related protein-1; DKK-1: Dickkopf-1; LRP 5/6: lipoprotein receptor-related proteins (LRP) 5 and 6; COL1A1: collagen type 1 alpha 1 chain; COL3A1: collagen type 3 alpha 1 chain; TAB1: transforming growth factor beta activated protein kinase 1; TGFR1: transforming growth factor beta receptor 1; TGFR2: transforming growth factor beta receptor 2; Wnt: Wnt signaling pathway; alpha-SMA: alpha-smooth muscle actin; CTGF: connective tissue growth factor; TIMPs: tissue inhibitors of metalloproteinases; IL-13R alpha 1: interleukin-13 receptor alpha 1; IL-4R alpha: interleukin-4 receptor alpha; STAT6: signal transducer and activator of transcription 6.

**Table 1 biomedicines-09-01471-t001:** Profibrotic miRNAs involved in SSc pathogenesis.

miRNA	Expression	Tissue Specimen(s)	Target Gene(s)	Reference(s)
miR-21	Upregulated	Fibroblasts Skin Bleomycin-treated mice skin samples	SMAD7	Zhu et al. [34,57] Jafarinejad-Farsangi et al. [58]
miR-145	Upregulated	Fibroblasts TGF-β1-stimulated fibroblasts	KLF4	Ly et al. [59]
miR-92a	Upregulated	Fibroblasts Serum TGF-β-stimulated fibroblasts	MMP1	Sing et al. [60]
miR-202-3p	Upregulated	Fibroblasts Skin	MMP1	Zhou et al. [61]
miR-4458	Upregulated	Fibroblasts	Unknown	Nakayama et al. [62]
miR-155	Upregulated	Fibroblasts Skin Serum	CSNK1A1 SHIP1	Yan et al. [63] Christmann et al. [64] Artlett et al. [65]
miR-27a-3p	Upregulated	Fibroblasts Skin Serum	sFRP-1	Henderson et al. [66]
miR-33a-3p	Upregulated	Fibroblasts	DKK-1	Henderson et al. [67]

KLF4: Kruppel-like factor 4; MMP1: matrix metalloproteinase 1; sFRP-1: secreted frizzled-related protein-1; DKK-1: Dickkopf-1.

**Table 2 biomedicines-09-01471-t002:** Antifibrotic miRNAs involved in SSc pathogenesis.

miRNA	Expression	Tissue Specimen(s)	Target Gene(s)	Reference(s)
miR-145	Downregulated	Fibroblasts Skin	SMAD3	Zhu et al. [34]
miR-29b	Downregulated	Fibroblasts Skin	COL1A1	Zhu et al. [34]
miR-let-7a	Downregulated	Fibroblasts Skin Serum	Unknown	Makino et al. [74]
miR-29a	Downregulated	Fibroblasts Skin Bleomycin-treated mice skin samples	COL1A1 COL3A1 TAB1	Maurer et al. [75] Jafarinejad-Farsangi et al. [58] Ciechomska et al. [76]
miR-3606-3p	Downregulated	Fibroblasts Skin	TGFBR2	Shi et al. [73]
miR-18a	Downregulated	Fibroblasts	Unknown	Nakayama et al. [62]
miR-150	Downregulated	Fibroblasts	ITGB3	Honda et al. [77]
miR-30b	Downregulated	Skin Serum Experimental mouse model	Unknown	Tanaka et al. [78]
miR-135b	Downregulated	Fibroblasts Serum Monocytes	STAT6	O’Reilly et al. [79]
miR-16-5p	Downregulated	Fibroblasts Serum	NOTCH2	Yao et al. [84]

COL1A1: collagen type 1 alpha 1 chain; COL3A1: collagen type 3 alpha 1 chain; TAB1: transforming growth factor beta activated protein kinase 1; TGFBR2: transforming growth factor beta receptor 2.

**Table 3 biomedicines-09-01471-t003:** Apoptosis and miRNAs.

miRNAs	Expression	Tissue Sample(s)	Regulatory Effect	Consequence	Reference
miR-29a	Downregulated	Fibroblasts TGF-β-stimulated fibroblasts	Increased Bax:Bcl2 ratio	Proapoptotic	Jafarinejad-Farsangi et al. [86]
miR-21	Upregulated	Fibroblasts	Decreased Bax:Bcl2 ratio	Antiapoptotic	Jafarinejad-Farsangi et al. [87]

**Table 4 biomedicines-09-01471-t004:** Microangiopathy and miRNAs.

miRNA	Tissue Samples	Regulatory Effect	Reference
miR-193b	Fibroblasts Skin HPASMCs cultures	uPA expression	Iwamoto et al. [88]
miR-126	Fibroblasts Skin HUVECs	EGFL7 expression	Liakouli et al. [89]

uPA: urokinase-type plasminogen activator; EGFL7: epidermal growth factor like-domain.

**Table 5 biomedicines-09-01471-t005:** Immune dysfunction and miRNAs.

miRNA	Tissue Sample	Regulatory Effect(s)	Reference
miR-30a-3p	Fibroblasts	BAFF production and secretion	Alsaleh et al. [90]
miR-26a-2-3p	Monocytes	Regulation of IFN-stimulated genes	Ciechomska et al. [91]

## Data Availability

Not applicable.

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
