# Peer review of "Novel Concepts in Systemic Sclerosis Pathogenesis: Role for miRNAs"

_biomedicines, 2021, doi:10.3390/biomedicines9101471_

Round 1

Reviewer 1 Report

While I have no major concerns with this paper, a couple of aspects could be improved:

  • in my view, many paragraphs are too long and would benefit from additional subheadings
  • "Table 1. Fibrosis and miRNAs. Profibrotic miRNAs." Suggest changing the heading to: "Profibrotic miRNAs involved in SSc pathogenesis." or similarly.

  • "Table 2. outlines the chief characteristics of the antifibrotic miRNAs identified in patients with SSc." - change chief to a different word, such as "main". 

  •  "Table 2. Fibrosis and miRNAs. Antifibrotic miRNAs." Suggest changing to "ANtifibrotic miRNAs in SSc pathogenesis". 

  • The paper is mainly focused on miRNA in skin fibrosis. Nevertheless, it would be useful to add a paragraph (and maybe a figure) of miRNA involved in pulmonary fibrosis, which is the main driver of mortality in SSc patients and compare these data to other fibrotic disease, such as IPF or RA-ILD.
  •  

     In Addition, there is epigenetic evidence of organ-specific fibrosis, such as kidney, liver, heart. it would be useful to add this aspect. 

Author Response

Response to Reviewer 1 Comments

Point 1: In my view, many paragraphs are too long and would benefit from additional subheadings

Response 1: Each paragraph is a summary of an original article. Some paragraphs are indeed long due to the complexity of the mechanisms and research methodology of a given study. Dividing the information or shortening of the paragraphs will require exclusion of some of the data and this might create confusion among readers.

Point 2: "Table 1. Fibrosis and miRNAs. Profibrotic miRNAs." Suggest changing the heading to: "Profibrotic miRNAs involved in SSc pathogenesis." or similarly.

Table 2. outlines the chief characteristics of the antifibrotic miRNAs identified in patients with SSc." - change chief to a different word, such as "main". 

"Table 2. Fibrosis and miRNAs. Antifibrotic miRNAs." Suggest changing to "ANtifibrotic miRNAs in SSc pathogenesis"

Response 2: All recommended changes were fulfilled accordingly

Point 3:  The paper is mainly focused on miRNA in skin fibrosis. Nevertheless, it would be useful to add a paragraph (and maybe a figure) of miRNA involved in pulmonary fibrosis, which is the main driver of mortality in SSc patients and compare these data to other fibrotic disease, such as IPF or RA-ILD.

Response 3: A new heading (Role of miRNAs in SSc-interstitial lung disease (SSc-ILD) pathogenesis) and several paragraphs with respect to the role of miRNAs in SSc-ILD were introduced according to recommendations.

Point 4:  In Addition, there is epigenetic evidence of organ-specific fibrosis, such as kidney, liver, heart. it would be useful to add this aspect. 

Response 4: There is little evidence of organ-specific fibrosis in kidneys, liver o heart in SSc and this is mainly due to the difficulty of obtaining tissue samples. Ethical aspects is another concern and therefore most studies focused either on serum and skin samples or on experimental mouse models.

Reviewer 2 Report

The conclusion is concise and well written.

In depth collection of reported information in a logical but difficult to read fashion due to the nature of molecular biology.

Thought provoking paragraph is described in Future directions.

Misuse of definitions and choice of words. Recommend minor English language revisions (instead of "druggable" use target of drug therapy, instead of "protean" use protein biomarkers or other suggestions etc)

Important summary for the experts and beginners in the field of scleroderma  

Given the complexity of certain pathways, more depiction of cascade and interactions might be of benefit for the reader. Figures of mechanisms may reveal additional hidden connections.

Author Response

Response to Reviewer 2 Comments

Point 1: Misuse of definitions and choice of words. Recommend minor English language revisions (instead of "druggable" use target of drug therapy, instead of "protean" use protein biomarkers or other suggestions etc)

Response 1: Changes were made according to reviewer's recommendations, with the exception of "protean" which doesn't stand for protein biomarkers but is a synonim for "variable"

Point 2: Given the complexity of certain pathways, more depiction of cascade and interactions might be of benefit for the reader. Figures of mechanisms may reveal additional hidden connections.

Response 2: Figure 2 depicting the regulatory effects of various profibrotic and antifibrotic miRNAs involved in SSc tissue fibrosis was created according to recommendations.

Reviewer 3 Report

The authors presented a summary of the current data on the possible participation of mRNA in the partogenesis of systemic sclerosis. Particularly noteworthy is the introduction, which, together with the figure, present epigenetic mechanisms in a perfect and synthetic way. The work is extremely interesting, however, a few minor points require the authors' attention:

  • Please discuss the potential reasons for the differences between the results obtained by the different research groups cited in the paragraph
  • Please provide the methodology for retrieving information on miRNAs in systemic sclerosis or state in the work limitations the possibility of a potential bias
  • An additional table containing the relationship of selected miRNAs with individual organ complications in systemic sclerosis would be a good complement to the manuscript
  • As to summarize the current role of miRNAs in the pathogenesis of systemic sclerosis, there are few cited studies from the last years 2020 and 2021.

Author Response

Response to Reviewer 3 Comments

Point 1: Please discuss the potential reasons for the differences between the results obtained by the different research groups cited in the paragraph 

Response 1: a brief explanation for the heterogeneity of the results between studies is provided in the conclusion paragraph: " SSc heterogeneity, small cohorts, permissive inclusion criteria without a clear distinction in terms of disease severity and status together with scarce data regarding current treatments are just some of the culprits responsible for the discordance between reports"

Point 2: Please provide the methodology for retrieving information on miRNAs in systemic sclerosis or state in the work limitations the possibility of a potential bias

Response 2: Heading 7. Limitations was introduced in response to this recommendation

Point 3: An additional table containing the relationship of selected miRNAs with individual organ complications in systemic sclerosis would be a good complement to the manuscript

Response 3: It is difficult to create a table to cover this recommendation as most studies concerning miRNAs and thei roles in SSc pathogenesis focused on skin biopsie, fibroblast cell lines, serum and experimental models due to the difficulty in obtaining tissue samples. Instead a new heading (5. Role of miRNAs in SSc-interstitial lung disease (SSc-ILD) pathogenesis) was introduced in the manuscript.

Point 4: As to summarize the current role of miRNAs in the pathogenesis of systemic sclerosis, there are few cited studies from the last years 2020 and 2021.

Response 4: With the addition of the new heading (5. Role of miRNAs in SSc-interstitial lung disease (SSc-ILD) pathogenesis), new references from 2021 were added.